# Controllable Helico-Conical Beam Generated with the Bored Phase

Xuejuan Liu [1,2], Shuo Liu [3,*] and Shubo Cheng [3,*]

1   College of Physics and Engineering, Chengdu Normal University, Chengdu 611130, China
2   School of Healthcare Technology, Chengdu Neusoft University, Chengdu 611844, China
3   School of Physics and Optoelectronic Engineering, Yangtze University, Jingzhou 434023, China
*   Correspondence: 2021710175@yangtzeu.edu.cn (S.L.); shubocheng@yangtzeu.edu.cn (S.C.)

**Abstract:** A controllable helico-conical beam is proposed in this paper. The intensity patterns and the local spatial frequency of the controllable helico-conical beams in the focal region are analyzed in detail. The results show that the length of the helico-conical beams can be customized by the variable parameter $k$, and the angular dimension of the bored spiral trajectory is dependent on the proportion $k/l$. Moreover, the focal-field energy flow density and orbital angular momentum distributions of the controllable helico-conical beams are also analyzed. The proposed helico-conical beams with controllable lengths can be potentially applied in the field of optical guiding.

**Keywords:** helico-conical beam; bored phase; optical manipulation; focal field distributions

## 1. Introduction

Optical beams with orbital angular momentum (OAM), e.g., optical vortices, have been widely used in optical communications and optical manipulation [1–13]. The high-order Bessel-Gauss beam with the non-diffractive and self-reconstructing properties was used for the three-dimensional trapping of microparticles [10]. Recently, unconventional types of optical beams have been proposed and analyzed. For example, spiral-type beams can be generated utilizing the astigmatic transforms [12], superposition of diffraction-free beams [13], focusing of shifted vortex beams [14], spiral toroidal lens [15], and refractive twisted microaxicons [16]. A new kind of beam, generated with inseparable helical and conical phases, has been proven to have a spiral-like intensity profile at the focal plane. The beam may find application for optical guiding [17]. Subsequently, the phase structure and interference characteristics of the helico-conical optical beam have been also analyzed [18–20]. In addition, some experts have investigated the self-reconstruction property of the optical beam [21]. The helical vortex structures are also connected with the depolarization of the laser beams and pulses [22,23]. We modified the helico-conical optical beam by adding a power exponent [24,25]. The proposed beams have controllable openings along the intensity trajectory, which can be dependent on the power exponent. Nathaniel Hermosa et al. proposed a method to control the intensity patterns of the helical beam, i.e., modifying the phase by boring a hole at the center of the helical phase [26,27]. The results show that the area of the bored hole has a great influence on the intensity patterns of the beam. The proposed method can also be introduced into the helico-conical phases. With the advancement of nanotechnology, the modulated beam can potentially be applied in the laser-induced nano-joining of nanoscale materials and the generation of light-induced helical-structured materials [28,29]. In isotropic polymers, 3D chiral microstructures can be achieved under the illumination of the spiral lobes and chirality generated by the helical wavefronts [30].

In this paper, we customized the helico-conical phases to control the length of the helico-conical beams. The customized phases are obtained by eliminating the partial helico-conical phase which is restricted with a filter. We will analyze the intensity patterns of

the controllable helico-conical beams in the focal region theoretically and experimentally. Based on the local spatial frequency, we discuss the dependence of the focal field intensity distributions of the controllable helico-conical beams on the filter parameter $k$. We also analyze the properties of the beam utilizing the energy flow density. The controllable helico-conical beams will find application for the optical guiding and light-induced helical-structured materials.

## 2. Controllable Helico-Conical Beams

The inseparable helical and conical phase profiles of the helico-conical beam can be written as [17]

$$\psi(r,\theta) = l\theta(K - r/r_0) \tag{1}$$

where the parameter $K$ is equal to 0 or 1, $l$ is the topological charge which determines the number and direction of the spiral wavefront, $\theta$ is the azimuth angle ranging from 0 to $2\pi$ in the polar coordinate system, and $r_0$ is the normalization factor of the radial coordinate $r$. In this paper, the usable size of the spatial light modulator (SLM) is $1920 \times 1080$ pixels with a pixel pitch of 8 µm, so the value of the radius $r_0$ can be set as 4.32 mm and $r$ can range from 0 to 4.32 mm. The wavelength of the laser beam in the simulations and experiments is set as 532 nm.

Based on Equation (1), the focal-field distributions of the beams can be simulated by the Fourier transform. In this paper, to generate the controllable spiral intensity patterns, we customized the phase hologram generated with the phase function in Equation (1). The bored phase in Figure 1c,f was obtained by customizing a partial helico-conical region along the screw dislocation of the helico-conical phase, which can be expressed by a matrix in the simulations. The bored phase profile can be written as $\psi_{Bored} = \psi(r,\theta) \cdot F(r,\theta)$, where $F(r,\theta)$ is a filter function. The filter can be given by

$$F(r,\theta) = \begin{cases} 1 & 2\pi k \leq |\psi(r,\theta)| \leq 2\pi|l| \\ 0 & \text{else} \end{cases} \tag{2}$$

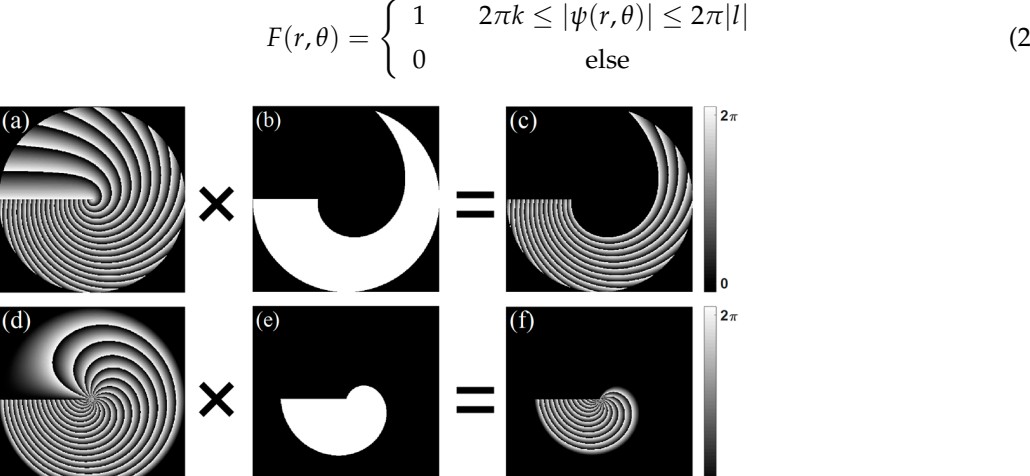

**Figure 1.** The generation of the bored helico-conical phase profiles. (**a,d**) the helico-conical phases with $l = 20$, $K = 0$, and $K = 1$, respectively. (**b,e**) the filters with $k = 6$, $l = 20$, $K = 0$, and $K = 1$, respectively. (**c,f**) the bored phases with $l = 20$, $k = 6$, $K = 0$, and $K = 1$, respectively.

Thus, the bored phase is also written as

$$\psi_{Bored} = \begin{cases} \psi(r,\theta) & 2\pi k \leq |\psi(r,\theta)| \leq 2\pi|l| \\ 0 & \text{else} \end{cases} \tag{3}$$

The helico-conical phases with $l = 20$, $K = 0$, and $K = 1$ are shown in Figure 1a,d respectively. Figure 1b,e show the filter $F$ with the parameters $l = 20$, $k = 6$, $K = 0$, and $K = 1$, respectively. Figure 1c,f show the corresponding bored helico-conical phase with $K = 0$ and $K = 1$, respectively. The variable parameter $k$ is an integer ranging from 0 to $l$. In fact, when the parameter $k$ is equal to 0, the bored phase profile is the whole helico-conical

phase. In the simulation and experiments, we eliminated the central intensity peak by adding the blazed gratings into the holograms in order to analyze the bored intensity distribution better.

## 3. Intensity Distributions at the Focal Plane

We analyzed the focal-field distributions of the controllable helico-conical beams. For ease of observation, we used normalized intensity for numerical simulations. The focal-field intensity distributions can be calculated by

$$u(\rho, \phi) = \int_0^{2\pi} \int_0^{\infty} F(r, \theta) \exp[i\psi(r, \theta)] \exp[-i2\pi r\rho \cos(\theta - \phi)] r dr d\theta \tag{4}$$

$F(r, \theta)$ is the filter function in Equation (2), and $\psi(r, \theta)$ is the phase function. It is convenient to evaluate Equation (4) numerically with the FFT algorithm. The intensity distribution can be calculated by $I(\rho, \phi) = |u(\rho, \phi)|^2$.

Figure 2a shows the helico-conical phase profile with $K = 0$ and $l = 20$. Figure 2b–e show the bored phase profiles with $K = 0$, $l = 20$, and $k = 1, 4, 7$, and 10, respectively. Figure 2f–j show the corresponding focal-field intensity distributions. Figure 3a shows the helico-conical phase profile with $K = 1$ and $l = 20$. Figure 3b–e show the bored phase profiles with $K = 1$, $l = 20$, and $k = 1, 4, 7$, and 10, respectively. Figure 3f–j show the corresponding focal-field intensity distributions. The heads of the helico-conical beams shown in Figures 2f and 3f can be marked with yellow arrows. The heads of the beams gradually vanish as the bored phase profile increases. It can also be seen from Figures 2 and 3 that the spiral intensity trajectories of the controllable helico-conical beams gradually become shorter with the increasing parameter $k$.

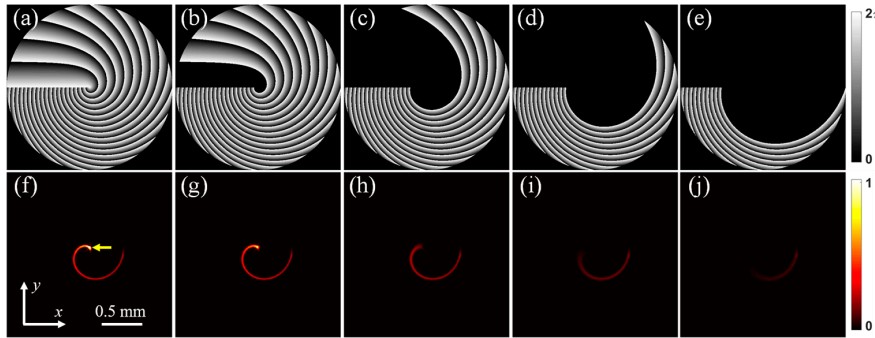

**Figure 2.** (**a**) The helico-conical phase profile with $K = 0$ and $l = 20$. (**b–e**) The bored phase profiles with $K = 0$, $l = 20$, and $k = 1, 4, 7$, and 10, respectively. (**f–j**) Simulated focal field intensity distributions of the controllable helico-conical beams with $K = 0$, $l = 20$, and $k = 0, 1, 4, 7$, and 10, respectively.

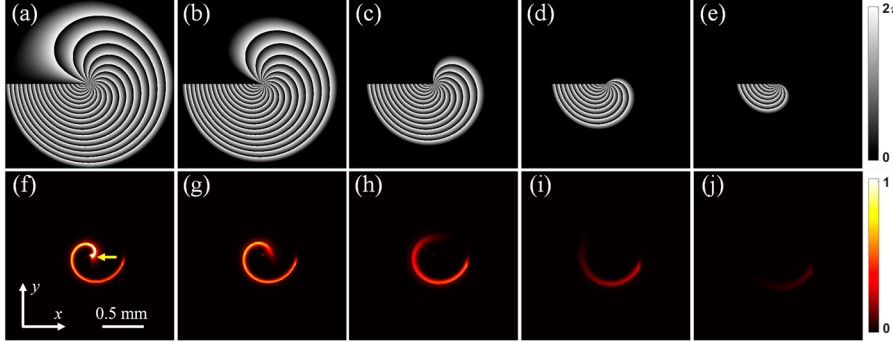

**Figure 3.** (**a**) The helico-conical phase profile with $K = 1$ and $l = 20$. (**b–e**) The bored phase profiles with $K = 1$, $l = 20$, and $k = 1, 4, 7$, and 10, respectively. (**f–j**) Simulated focal field intensity distributions of the controllable helico-conical beams with $K = 1$, $l = 20$, and $k = 0, 1, 4, 7$, and 10, respectively.

In this paper, the local spatial frequency distribution was used to analyze the spiral intensity trajectories of the controllable helico-conical beams. Generally, the approximate mapping of the local spatial frequency can be expressed by [17]

$$\xi' = \frac{1}{2\pi}\frac{\partial}{\partial x}\overline{\psi}(x,y) \quad \text{and} \quad \zeta' = \frac{1}{2\pi}\frac{\partial}{\partial y}\overline{\psi}(x,y) \tag{5}$$

Thus, the local spatial frequency of the controllable helico-conical beams can be described by the equation $\xi' = \xi \cdot F$ and $\zeta' = \zeta \cdot F$, respectively [12]. Based on the phase function in Equation (1) and the filter function $F$ in Equation (2), the approximate mapping of the local spatial frequency for the beams with $K = 0$ and $K = 1$ in polar coordinates are expressed as Equations (6) and (7), respectively.

$$\xi'_{K=0} = \begin{cases} \frac{l}{2\pi r_0}(-\theta\cos\theta + \sin\theta) & 2\pi k \leq |\psi(r,\theta)| \leq 2\pi|l| \\ 0 & \text{else} \end{cases}$$
$$\text{and} \tag{6}$$
$$\zeta'_{K=0} = \begin{cases} \frac{l}{2\pi r_0}(-\theta\sin\theta - \cos\theta) & 2\pi k \leq |\psi(r,\theta)| \leq 2\pi|l| \\ 0 & \text{else} \end{cases}$$

$$\xi'_{K=1} = \begin{cases} \frac{l}{2\pi r_0}(-\theta\cos\theta - \frac{r_0-r}{r}\sin\theta) & 2\pi k \leq |\psi(r,\theta)| \leq 2\pi|l| \\ 0 & \text{else} \end{cases}$$
$$\text{and} \tag{7}$$
$$\zeta'_{K=1} = \begin{cases} \frac{l}{2\pi r_0}(-\theta\sin\theta + \frac{r_0-r}{r}\cos\theta) & 2\pi k \leq |\psi(r,\theta)| \leq 2\pi|l| \\ 0 & \text{else} \end{cases}$$

The plots ($\xi'_{K=0}$, $\zeta'_{K=0}$) of the controllable helico-conical beams with $K = 0$, $l = 20$, and $k = 0, 1, 4, 7$, and 10 are shown in Figure 4a–e, respectively. The plots ($\xi'_{K=1}$, $\zeta'_{K=1}$) of the controllable helico-conical beams with $K = 1$, $l = 20$, and $k = 0, 1, 4, 7$, and 10 are shown in Figure 4f–j, respectively. In Figure 4a, the points accumulate in a spiral, corresponding to the helico-conical beam with $K = 0$ and $l = 20$. With $k$ increasing, the partial region of the phase was bored, as is shown in Figure 2b–e. The resulting spiral intensity trajectories of the beams became shorter. The inner and outer phases of the helico-conical phase with $K = 0$ corresponded to the head and tail of the spiral intensity trajectories, respectively. Contrariwise, the inner and outer phases of the helico-conical phase with $K = 1$ corresponded to the tail and head of the spiral intensity trajectories, respectively. From Figure 4b,g, it also can be seen that an observable dislocation of the spiral's head occurs. The spot diagrams of the local spatial frequency shown in Figure 4a–j agree well with the results shown in Figures 2f–j and 3f–j, respectively.

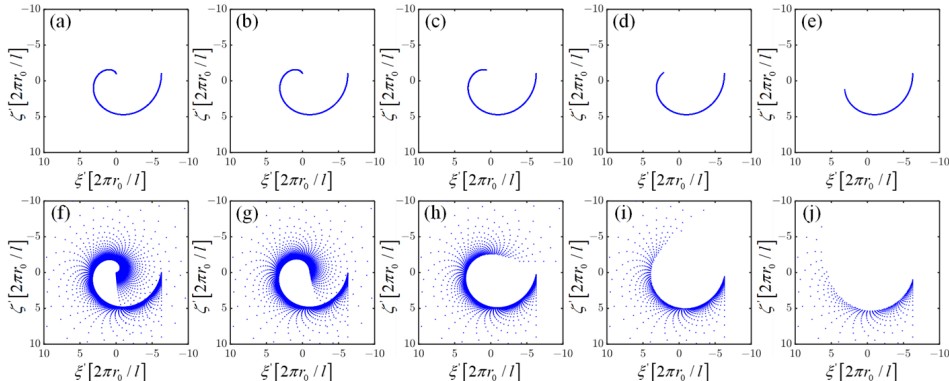

**Figure 4.** (**a–e**) Spot diagrams of the local spatial frequency ($\xi'_{K=0}$, $\zeta'_{K=0}$) for the controllable helico-conical beams with $K = 0$, $l = 20$, and $k = 0, 1, 4, 7$, and 10, respectively. (**f–j**) Spot diagrams of the local spatial frequency ($\xi'_{K=1}$, $\zeta'_{K=1}$) for the beams with $K = 1$, $l = 20$, and $k = 0, 1, 4, 7$, and 10, respectively.

The dependence of the focal-field intensity for the controllable helico-conical beams on the parameter $k$ is analyzed in this paper. The length of the spiral intensity trajectories shown in Figure 4 cannot be measured simply and directly. Thus, we used the angular dimension to describe the change of the spiral intensity trajectories indirectly. The spiral's head was treated as the origin, and the endpoint along the spiral trajectory was treated as the tail. The azimuth between the head-to-tail connecting line and the horizontal axis (the $x$-axis in Figure 5) describes the length of the bored spiral intensity trajectories. The calculation of the azimuth is shown in Figure 5. In the spot diagrams of the local spatial frequency for the controllable beams with $K = 0$, $l = 20$, and $k = 0$ (see Figure 5a), the corresponding azimuth for the whole helico-conical beam is $3\pi/2$. As an example, the corresponding azimuth for the controllable helico-conical beams with $K = 0$, $l = 20$, and $k = 12$ is about $0.6712\pi$, as is shown in Figure 5b.

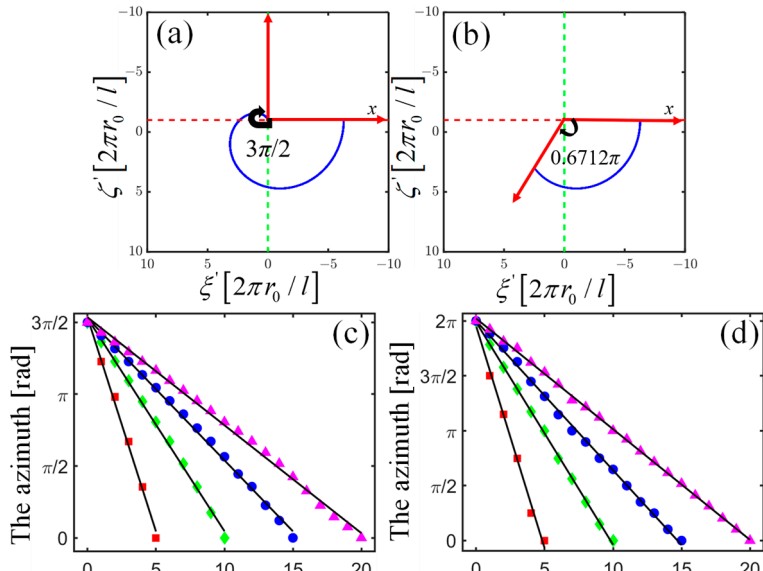

**Figure 5.** (**a**) The azimuth corresponding to the whole helico-conical beam is $3\pi/2$. (**b**) The azimuth corresponding to the controllable helico-conical beam with $K = 0$, $l = 20$, and $k = 12$ is about $0.6712\pi$. (**c**) Plots of the azimuth versus the parameter $k$ of the controllable helico-conical beams with $K = 0$, $l = 5, 10, 15$, and 20, respectively. (**d**) Plots of the azimuth versus the parameter $k$ of the controllable helico-conical beams with $K = 1$, $l = 5, 10, 15$, and 20, respectively. The lines represent the linear regression lines.

With the method shown in Figure 5a,b, the plots of the azimuth versus the parameter $k$ for the controllable helico-conical beams with $K = 0$ and $K = 1$ are shown in Figure 5c,d respectively. The azimuth calculated with the local spatial spectrum corresponding to the different topological charges $l = 5, 10, 15, 20$ is marked with different legends and colors. The linear fitting of the data was implemented, and the corresponding linear regression lines are marked with the corresponding colors. The results demonstrate that the azimuth scales linearly with the parameter $k$ and the slope of the linear regression line can be dependent on the topological charge $l$. The dependence of the azimuth $\varphi$ on the parameter $k$ can be empirically written as

$$\varphi(k) = \begin{cases} \frac{3\pi/2}{l}(l-k) & K = 0 \\ \frac{2\pi}{l}(l-k) & K = 1 \end{cases} \tag{8}$$

In Equation (6), the magnitude of the slope is $-\frac{3\pi/2}{l}$ and $-\frac{2\pi}{l}$, respectively. The minus sign before the parameter $k$ indicates that the angular dimension decreases as the variable parameter $k$ increases. Thus, we can generate customizable helico-conical beams with the bored helico-conical phases.

We also analyzed the focal-field intensity distributions with the Poynting vector. The energy flow in the focal-field region of the controllable helico-conical beams was calculated. The Poynting vector can be described by the following formula [1],

$$
\begin{aligned}
\left\langle \vec{S} \right\rangle &= \frac{c}{4\pi} \left\langle \vec{E} \times \vec{B} \right\rangle \\
&= \frac{c}{8\pi} [(E^* \times B) + (E \times B^*)] \\
&= \left[ \frac{i\omega c}{8\pi} \left( u \frac{\partial u^*}{\partial y} - u^* \frac{\partial u}{\partial y} \right) \right] \vec{x} + \left[ \frac{i\omega c}{8\pi} \left( u \frac{\partial u^*}{\partial x} - u^* \frac{\partial u}{\partial x} \right) \right] \vec{y} + \frac{\omega k c}{4\pi} |u|^2 \vec{z}
\end{aligned}
\tag{9}
$$

where $c$, $\vec{E}$ and $\vec{B}$ represent light speed, electric field intensity vector, and magnetic field intensity vector, respectively; $\vec{x}$, $\vec{y}$, and $\vec{z}$ denote the unit vectors along the $x$, $y$, and $z$ directions, respectively. $\omega$ and $k$ ($k = 2\pi/\lambda$) are the angular frequency and the wave number of the controllable helico-conical beam, respectively. The three components in the $x$, $y$, and $z$ directions denote the energy flow in the different directions, respectively.

The transversal energy flows (i.e., $x$-$y$ plane) of the controllable helico-conical beams were calculated and analyzed in this paper. Figure 6a–j show the focal-field transverse energy flow of the controllable helico-conical with $K = 0$, $l = 20$, and $k = 0, 1, 4, 7$, and 10; $K = 1$, $l = 20$, and $k = 0, 1, 4, 7$, and 10, respectively. The direction and magnitude of the green arrows in the figures demonstrate the direction and size of the energy flow at the Fourier transform plane. From the figures, it can be seen that the energy flow is flowing towards the interior of the bored helico-conical beam, which demonstrated that the beam can trap particles and bind particles towards the spiral trajectories.

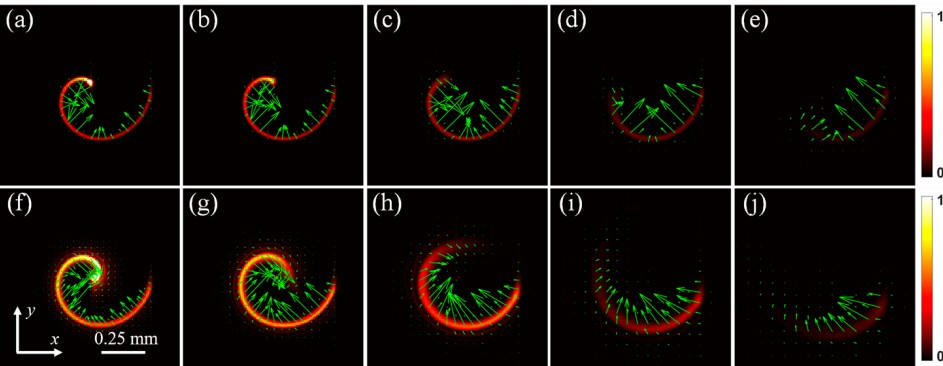

**Figure 6.** (**a**–**e**) The transverse energy flow of the controllable helicon-conical beams with $K = 0$, $l = 20$, and $k = 0, 1, 4, 7$, and 10, respectively. (**f**–**j**) The transverse energy flow of the controllable helicon-conical beams with $K = 1$, $l = 20$, and $k = 0, 1, 4, 7$, and 10, respectively.

The controllable helico-conical beams also carry the OAM at the focal plane. The focal-field OAM density of the controllable helico-conical beams in free space can be described by

$$
j_z = (r \times \varepsilon_0 \langle E \times B \rangle)_z = x S_y - y S_x
\tag{10}
$$

where, $E$ and $B$ denote the electric and magnetic fields, respectively; $r = (x^2 + y^2)^{1/2}$, $S_x$ and $S_y$ are the Poynting vector along the $x$ and $y$ directions, respectively.

Figure 7a–e demonstrate the focal-field OAM density distributions of the controllable helico-conical beams with $K = 0$, $l = 20$, and $k = 0, 1, 4, 7$, and 10, respectively. Figure 7f–j illustrate the focal-field OAM density distributions of the controllable helico-conical beam with $K = 1$, $l = 20$, and $k = 0, 1, 4, 7$, and 10, respectively. The OAM density distributions of the controllable beams at the focal plane are roughly consistent with the focal-field intensity distributions. With the missing part of the helico-conical phase profile increased, the OAM density distributions are totally changed and follow the bored spiral lines. The OAM density also decreased gradually. We analyzed the relationship between the normalized OAM density and the parameter $k$. The plots of the normalized OAM density versus

the parameter $k$ are shown in Figure 8. When $K = 1$, the focal-field OAM density of the controllable helico-conical beam is relatively large. However, as the parameter $k$ increased, the OAM density decreased. When the parameter $k$ is larger (e.g., $k = 20$), the whole helico-conical phase profile is missing basically and the corresponding OAM density is close to 0. Thus, the helico-conical beam can be modulated by considering the above results and can be applied to optical trapping and light-induced helical-structured materials.

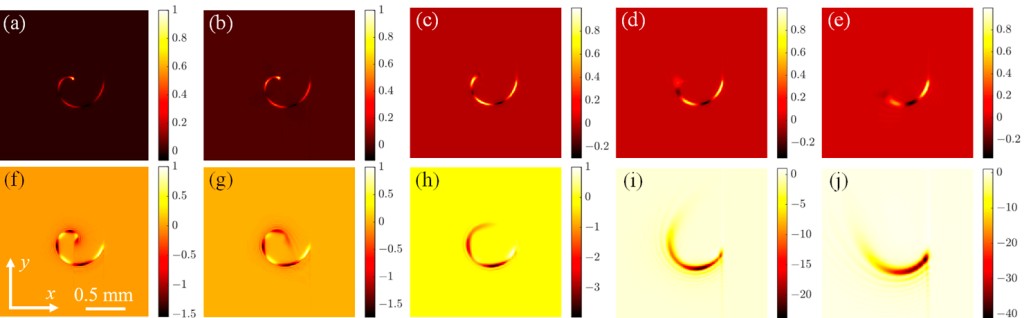

**Figure 7.** (**a**–**e**) The focal-field OAM density distributions of the controllable helico-conical beams with $K = 0$, $l = 20$, and $k = 0$, 1, 4, 7, and 10, respectively. (**f**–**j**) The focal-field OAM distributions of the controllable helico-conical beams with $K = 1$, $l = 20$, and $k = 0$, 1, 4, 7, and 10, respectively.

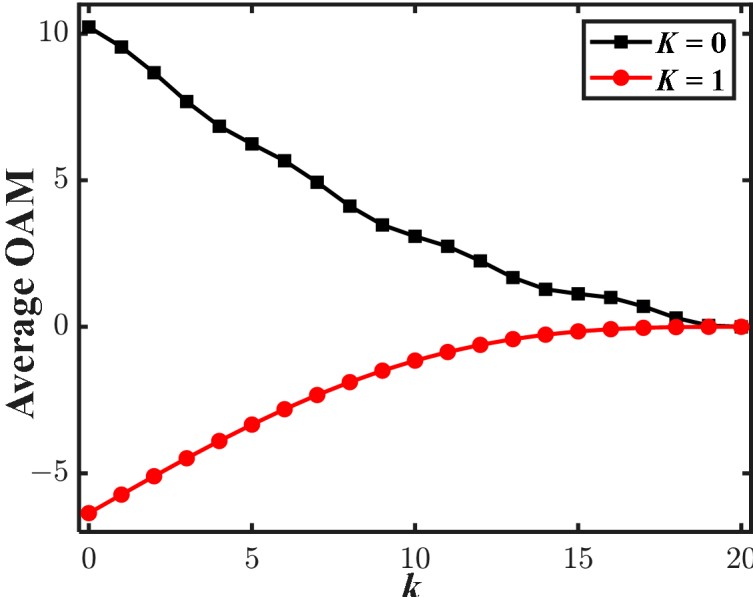

**Figure 8.** Influence of $k$ and $K$ on the OAM density of the controllable helico-conical beams.

In this paper, the experimental focal field intensity patterns of the controllable helico-conical beams have been analyzed. Figure 9a shows the schematic setup for generating the controllable helico-conical beams. A collimated and expanded laser ($\lambda = 532$ nm) beam impinged onto the SLM (reflective type), which was encoded with a computer-generated hologram. The flat convex lens $L_1$ ($f_1 = 30$ mm) and $L_2$ ($f_2 = 200$ mm) was used for the collimation and expansion of the optical beams. The lens ($L_3$, $f_3 = 150$ mm) was a Fourier-transform lens. The intensity cross-sections of the desired beams were captured by the CCD camera. During the experiments, the corresponding phase profiles in Figures 2a–e and 3a–e were loaded on the SLM, respectively. The corresponding intensity patterns at the focal plane are shown in Figure 9b–k, respectively. The experiment results are consistent with the simulated ones shown in Figures 2f–j and 3f–j, respectively. The focal field intensity distributions were determined by the parameter $k$.

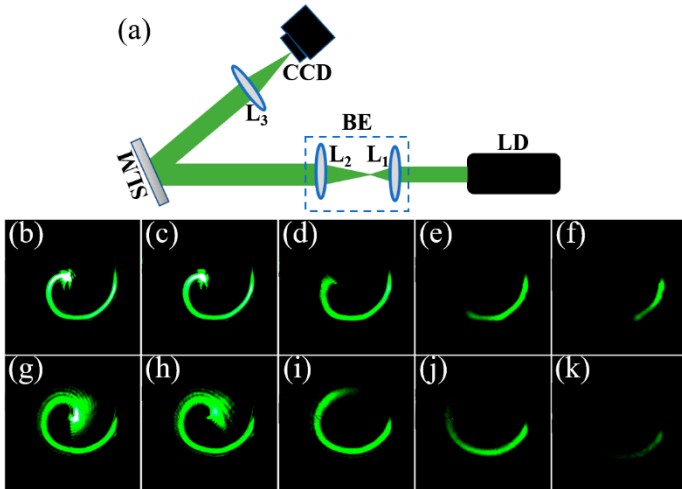

**Figure 9.** (**a**) The schematic experimental setup for generating the controllable helico-conical beams. (**b**–**f**) The focal field intensity distributions of the controllable helico-conical beams with $K = 0$, $l = 20$, and $k = 0, 1, 4, 7$, and 10, respectively. (**g**–**k**) The focal field intensity distributions of the controllable helico-conical beams with $K = 1$, $l = 20$, and $k = 0, 1, 4, 7$, and 10, respectively.

## 4. Conclusions

　　A controllable helico-conical beam with a different intensity trajectory was proposed in this paper. The focal-field intensity distributions of the controllable helico-conical beams have been discussed. The spiral intensity trajectories of the controllable helico-conical beams were shorter with the increasing parameter $k$. The resulting azimuth scales linearly with the parameter $k$. The arrows of the energy flow density point to the interior of the spiral-like beam. When the missing part of the helico-conical phase profile increases, the OAM density distributions are totally changed, and follow the bored spiral lines. The OAM density also decreases gradually. The proposed helico-conical beams with the controllable length are promising in the optical guiding of microparticles and light-induced helical-structured materials.

**Author Contributions:** Conceptualization, S.L., X.L. and S.C.; data curation, X.L.; formal analysis, X.L. and S.C.; methodology, S.L. and X.L.; resources, X.L.; software, X.L.; data curation, S.L.; writing-original draft preparation, S.L. and X.L.; writing-review and editing, X.L. and S.C. All authors have read and agreed to the published version of the manuscript.

**Funding:** The authors are grateful for the support of the National Natural Science Foundation of China (No. 11904032) and the Talent Introduction Program of Chengdu Normal University (No. YJRC2021-14).

**Institutional Review Board Statement:** Not applicable.

**Informed Consent Statement:** Not applicable.

**Data Availability Statement:** Not applicable.

**Conflicts of Interest:** The authors declare no conflict of interest.

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
