# Peer review of "Controllable Helico-Conical Beam Generated with the Bored Phase"

_photonics, doi:10.3390/photonics10050577_

Round 1

Reviewer 1 Report

The authors have written an interesting paper on the properties and generation of exotic oam beams (helical - conical). I think that the work is quite worthy of publication in Photonics, but it seems to me that it needs to be improved a little. Here are my comments: 

1. It seems to me that for people who are not fully involved in the topic of such works, it is necessary to explain the physical meaning of the parameters that are taken from reference 12 (lines 108 and 109) and used in formulas 4 and 5. This will greatly enhance the understanding of this work.

2. The same can be said about the formula for the Poynting vector in formula 7, where the notation u and the conjugate u are used. At the same time, it is not explained that this is possible just a function describing the electric field of the beam in the paraxial approximation from reference 1 as far I understand.

3. And finally, it would be good if the authors would tell a little more about the possible practical applications of these exotic beams. It is thanks to their interesting properties and complex structure.

Author Response

Replies to Reviewer #1:

General Remarks: The authors have written an interesting paper on the properties and generation of exotic OAM beams (helical-conical). I think that the work is quite worthy of publication in Photonics, but it seems to me that it needs to be improved a little. Here are my comments:

Response: Thanks for your appreciation of our contributions.

Comment 1: It seems to me that for people who are not fully involved in the topic of such works, it is necessary to explain the physical meaning of the parameters that are taken from reference 12 (lines 108 and 109) and used in formulas 4 and 5. This will greatly enhance the understanding of this work.

Response 1: Thanks for your comments.

 The local spatial frequency can be approximated as

According to the conversion relationship between Cartesian coordinates and polar coordinates, it is easy to get:

Based on the formula above, the local spatial frequency can be calculated.

So, the sentences “Generally, the approximate mapping of the local spatial frequency can be expressed by [17]

                                                                        (5)

” should be added in the first paragraph of page 4.

Comment 2: The same can be said about the formula for the Poynting vector in formula 7, where the notation u and the conjugate u are used. At the same time, it is not explained that this is possible just a function describing the electric field of the beam in the paraxial approximation from reference 1 as far I understand.

Response 2: Thanks for your suggestions.

u is the calculated amplitude by the Fourier transform based on the angular spectrum theory. It can be expressed by:

                                                                       (4)

 is the filter function in Eq. (2), and  is the phase function. It is convenient to evaluate Eq. (4) numerically with FFT algorithm. And the intensity distribution can be calculated by .

So, the sentence “The focal-field intensity distributions can be calculated by

F(r, θ) is the filter function, and  is the phase function. It is convenient to evaluate Eq. (4) numerically with FFT algorithm. The intensity distribution can be calculated by .” can be added in the first paragraph of page 3.

Comment 3: And finally, it would be good if the authors would tell a little more about the possible practical applications of these exotic beams. It is thanks to their interesting properties and complex structure.

Response 3: Thanks for your comments.

The exotic beams with the controllable length will be mainly promising in optical manipulation of microparticles and the light-induced helical-structured materials.

Reviewer 2 Report

In their manuscript “Controllable helico-conical beam generated with the bored phase” the authors introduce a method to modify the length of the tail in helico-conical beams by changing the size of the central aperture in the phase profile. They explain their approach and present mostly numerical simulations and some experimental intensity images.

While I believe the work is interesting and suitable for Photonics, I would like the authors to address the following comments before I recommend it for publication.

1.     Looking at the intensity images, particularly in Figures 2 and 3, I would like to know if the power of the beam is the same in all simulations. First, since there is no scale bar, I cannot know if the scale is the same for all images (or not), thus making it impossible to compare the images on a fair basis. Second, assuming all scales are the same, why is the case k=4 the brightest? I’d have imagined that if the intensity of the incoming beam is constant across all the SLM the brightest should be k=0, which is the case in the experimental images (Fig. 10). Why this is not the case in Figures 2 and 3?

2.     The authors state that “The length of the spiral intensity trajectories shown in Figure 4 cannot be measured directly.” Please elaborate. Why cannot the length be measured?

3.     Referring to Fig. 7, the authors state “From the figures it can be seen that the energy flow is flowing towards the head of the bored helico-conical beam.” However, some figures don’t seem to follow this. Specifically, in Fig. 7f the arrows point left-upward, but from Fig. 3j it seems the head of the beam is on the right, thus contradicting the statement. I recommend the authors overlap the arrows (with much less density) onto the beam intensity to facilitate the analysis.

4.     Regarding the OAM density, what do the authors mean by “have some discontinuities”? In the images, it is clearly seen the density varies along the beam, but a discontinuity is not recognizable. A scale bar is necessary, especially considering the OAM can take positive and negative values. The color map doesn’t help either.

5.     The authors write “The plots of the obtained OAM density versus the parameter k are shown in Figure 9.” However, since the OAM density is a local measure, what are you plotting on Fig. 9? Is it the integrated OAM density? Or the OAM density at a specific point?

6.     Why are the behaviors of the OAM for K=0 and K=1 different? Please comment

7.     In the experiment, the authors use a 15 cm lens to Fourier transform the SLM plane, which seems very sort. What is the angle between the incoming beam and the SLM? Is that within the recommended range?

8.     In the conclusions, the authors state “As the parameter k increased, the focal-field OAM distributions of the controllable helico-conical beams become shorter and shorter.” Do they mean the OAM content decreases as k increases? Please rephrase.

9.     There are too many figures. Consider reasonably grouping them. For instance, the diagrams shown in Fig. 5 could be incorporated into Fig. 6.

10.  Data points in Fig. 6 are hardly visible.

11.  The arrows in Fig. 7 are too thin, too close to each other with poor resolution and overlapping, making it difficult to see patterns.

English is fine but could be improved. Some sentences are difficult to understand, and there are few words used unusually.

Author Response

Replies to Reviewer #2:

General Remarks: In their manuscript “Controllable helico-conical beam generated with the bored phase” the authors introduce a method to modify the length of the tail in helico-conical beams by changing the size of the central aperture in the phase profile. They explain their approach and present mostly numerical simulations and some experimental intensity images.

While I believe the work is interesting and suitable for Photonics, I would like the authors to address the following comments before I recommend it for publication.

Response: Thank you for your appreciation of our contributions.

Comment 1: Looking at the intensity images, particularly in Figures 2 and 3, I would like to know if the power of the beam is the same in all simulations. First, since there is no scale bar, I cannot know if the scale is the same for all images (or not), thus making it impossible to compare the images on a fair basis. Second, assuming all scales are the same, why is the case k = 4 the brightest? I’d have imagined that if the intensity of the incoming beam is constant across all the SLM the brightest should be k=0, which is the case in the experimental images (Fig. 10). Why this is not the case in Figures 2 and 3?

Response 1: Thanks for your suggestions. The normalized intensity is confused in the previous Figures 2 and 3. In the simulations, we normalized the focal-field intensity again. The new results are consistent with the experimental ones.

The color bar and scale bar are added into Figures 2 and 3, and the revised figures are as follows.

Figure 2. (a) The helico-conical phase profile with K = 0, l = 20. (b-e) The bored phase profiles with K = 0, l = 20, k = 1, 4, 7, and 10, respectively. (f-j) Simulated focal field intensity distributions of the controllable helico-conical beams with K = 0, l = 20, k = 0, 1, 4, 7, and 10, respectively.

Figure 3. (a) The helico-conical phase profile with K = 1, l = 20. (b-e) The bored phase profiles with K = 1, l = 20, k = 1, 4, 7, and 10, respectively. (f-j) Simulated focal-field intensity distributions of the controllable helico-conical beams with K = 1, l = 20, k = 0, 1, 4, 7, and 10, respectively.

Comment 2: The authors state that “The length of the spiral intensity trajectories shown in Figure 4 cannot be measured directly.” Please elaborate. Why cannot the length be measured?

Response 2: Thank you for your suggestions.

I am sorry to confuse you. The length of the spiral intensity trajectories shown in Figure 4 can be measured mathematically and theoretically. In fact, we have not found a method of measuring the spiral intensity trajectories length directly, so we use the angular dimension to describe the length of the spiral intensity trajectories indirectly in this paper.

So the sentence “The length of the spiral intensity trajectories shown in Figure 4 cannot be measured directly.” in the first paragraph of page 5 can be revised as “The length of the spiral intensity trajectories shown in Figure 4 cannot be measured simply and directly. Thus, we use the angular dimension to describe the change of the spiral intensity trajectories indirectly.”

Comment 3: Referring to Fig. 7, the authors state “From the figures it can be seen that the energy flow is flowing towards the head of the bored helico-conical beam.” However, some figures don’t seem to follow this. Specifically, in Fig. 7f the arrows point left-upward, but from Fig. 3j it seems the head of the beam is on the right, thus contradicting the statement. I recommend the authors overlap the arrows (with much less density) onto the beam intensity to facilitate the analysis.

Response 3: Thank you for your suggestions.

I am sorry to confuse you. In fact, the energy flow is flowing towards the interior of the bored helico-conical beam. The heads of the bored helico-conical beams shown in Figure 2f and Figure 3f are on the right and can be marked with the arrows. The beam heads will vanish gradually with the bored phase profile increased.

We have overlapped the arrows (with much less density) onto the beam intensity, as is shown in the revised Figure 6.

The sentences “The heads of the helico-conical beams shown in Figure 2f and Figure 3f can be marked with the arrows. The heads of the beams will gradually vanish with the bored phase profile increased.” can be added into the second paragraph of page 3.

The sentence “From the figures it can be seen that the energy flow is flowing towards the head of the bored helico-conical beam, which proves that the intensity of the head for the bored beam is greater and the beam has the ability to trap and manipulate microparticles.” in the second paragraph of page 6 can be revised as “From the figures it can be seen that the energy flow is flowing towards the interior of the bored helico-conical beam, which demonstrated that the beam can trap particles and bind particles towards the spiral trajectories.”.

The revised Figure 7 is as follows.

Figure 6 (a-e) The transverse energy flow of the controllable helico-conical beams with K = 0, l = 20, k = 0, 1, 4, 7, and 10, respectively. (f-j) The transverse energy flow of the controllable helicon-conical beams with K = 1, l = 20, k = 0, 1, 4, 7, and 10, respectively.

Comment 4: Regarding the OAM density, what do the authors mean by “have some discontinuities”? In the images, it is clearly seen the density varies along the beam, but a discontinuity is not recognizable. A scale bar is necessary, especially considering the OAM can take positive and negative values. The color map doesn’t help either.

Response 4: Thank you for your suggestions

We re-analyzed the OAM density distribution of the controllable helico-conical beam, and found that our statement in the origin manuscript was wrong. The orbital angular momentum is indeed positive or negative.

The sentence “The actual focal-field intensity distributions are continuous [see Figures 2f-2j and Figures 3f-3j], but the obtained OAM density distributions according to the Pointing vector have some discontinuities.” in the second paragraph of page 7 should be deleted.

The sentence in the second paragraph of page 7 “The OAM density distributions of the controllable beams at the focal plane are roughly consistent with the focal-field intensity distribution and the OAM density distributions become shorter as the variable parameter k increases.”  is revised as “The OAM density distributions of the controllable beams at the focal plane are roughly consistent with the focal-field intensity distribution. With the missing part of the helico-conical phase profile increased, the OAM density distributions are totally changed, and follow the bored spiral lines. The OAM density will be also decreased gradually.”

The sentences in the second paragraph of page 8 “When K = 1, the focal-field OAM density of the controllable helico-conical beam is relatively large and decreased rapidly with the parameter k increased. But the focal-field OAM density of the controllable helico-conical beam with K = 0 is small and decreased slowly with the increasing parameter k.” are revised as “When K = 1, the focal-field OAM density of the controllable helico-conical beam is relatively large. But the OAM density will be decreased with the parameter k increased. When the parameter k is lager (e.g., k=20), the whole helico-conical phase profile is missing basically and the corresponding OAM density is close to 0.”

The scale bars have been added into Figure 7. The revised Figure 7 is as follows.

Figure 7. (a-e) The focal-field OAM distributions of the controllable helico-conical beams with K = 0, l = 20, k = 0, 1, 4, 7, and 10, respectively. (f-j) The focal-field OAM distributions of the controllable helico-conical beams with K = 1, l = 20, k = 0, 1, 4, 7, and 10, respectively.

Comment 5: The authors write “The plots of the obtained OAM density versus the parameter k are shown in Figure 9.” However, since the OAM density is a local measure, what are you plotting on Fig. 9? Is it the integrated OAM density? Or the OAM density at a specific point?

Response 5: Thank you for your suggestions. What we discussed in the manuscript is the integrated OAM density.

With the missing part of the helico-conical phase profile increased, the OAM density distributions are totally changed, and follow the spiral lines. The OAM density will be also decreased gradually.

The sentence “The OAM density distributions of the controllable beams at the focal plane are roughly consistent with the focal-field intensity distribution and the OAM density distributions become shorter as the variable parameter k increases.” can be revised as “The OAM density distributions of the controllable beams at the focal plane are roughly consistent with the focal-field intensity distribution. With the missing part of the helico-conical phase profile increased, the OAM density distributions are totally changed, and follow the bored spiral lines. The OAM density will be also decreased gradually.”

The sentences in the second paragraph of page 8 “When K = 1, the focal-field OAM density of the controllable helico-conical beam is relatively large and decreased rapidly with the parameter k increased. But the focal-field OAM density of the controllable helico-conical beam with K = 0 is small and decreased slowly with the increasing parameter k.” are revised as “When K = 1, the focal-field OAM density of the controllable helico-conical beam is relatively large. But the OAM density will be decreased with the parameter k increased. When the parameter k is lager (e.g., k=20), the whole helico-conical phase profile is missing basically and the corresponding OAM density is close to 0.”

The revised Figure is as follows.

Figure 8 Influence of k and K on the OAM density of the controllable helico-conical beams.

Comment 6: Why are the behaviors of the OAM for K = 0 and K = 1 different? Please comment

Response 6: Thank you for your suggestions.

From Figures 2 and 3, it can be seen that the phase profiles of the helicon-conical beams with K = 0 and K = 1 are different. The value of the calculated Poynting vector with Equation (9) is also different. The corresponding behaviors of the OAM for K = 0 and K = 1 can be different. The revised Figure 7 is as follows.

Figure 7. (a-e) The focal-field OAM distributions of the controllable helico-conical beams with K = 0, l = 20, k = 0, 1, 4, 7, and 10, respectively. (f-j) The focal-field OAM distributions of the controllable helico-conical beams with K = 1, l = 20, k = 0, 1, 4, 7, and 10, respectively.

Comment 7: In the experiment, the authors use a 15 cm lens to Fourier transform the SLM plane, which seems very short. What is the angle between the incoming beam and the SLM? Is that within the recommended range?

Response 7: Thank you for your suggestions.

The angle between the incoming beam and the SLM is about 15-20 degrees.

Comment 8: In the conclusions, the authors state “As the parameter k increased, the focal-field OAM distributions of the controllable helico-conical beams become shorter and shorter.” Do they mean the OAM content decreases as k increases? Please rephrase.

Response 8: Thanks for your suggestions. I am sorry for our expressions.

The sentence in the conclusions “As the parameter k increased, the focal-field OAM distributions of the controllable helico-conical beams become shorter and shorter.” is revised as “With the missing part of the helico-conical phase profile increased, the OAM density distributions are totally changed, and follow the bored spiral lines. The OAM density will be also decreased gradually.”.

Comment 9: There are too many figures. Consider reasonably grouping them. For instance, the diagrams shown in Fig. 5 could be incorporated into Fig. 6.

Response 9: Thank you for your suggestions.

We have combined Figure 5 and Figure 6.

Figure 5. (a) The azimuth corresponding to the whole helico-conical beam is 3π/2. (b) The azimuth corresponding to the controllable helico-conical beam with K = 0, l = 20, and k = 12 is about 0.6712π. (c) Plots of the azimuth versus the parameter k of the controllable helico-conical beams with K = 0, l = 5, 10, 15, and 20, respectively. (d) Plots of the azimuth versus the parameter k of the controllable helico-conical beams with K = 1, l = 5, 10, 15, and 20, respectively. The lines represent the linear regression lines.

Comment 10: Data points in Fig. 6 are hardly visible.

Response 10: Thank you for your suggestions.

Figure 6 is revised and combined with Figure 5.

Figure 5 (a) The azimuth corresponding to the whole helico-conical beam is 3π/2. (b) The azimuth corresponding to the controllable helico-conical beam with K = 0, l = 20, and k = 12 is about 0.6712π. (c) Plots of the azimuth versus the parameter k of the controllable helico-conical beams with K = 0, l = 5, 10, 15, and 20, respectively. (d) Plots of the azimuth versus the parameter k of the controllable helico-conical beams with K = 1, l = 5, 10, 15, and 20, respectively.  The lines represent the linear regression lines.

Comment 11: The arrows in Fig. 7 are too thin, too close to each other with poor resolution and overlapping, making it difficult to see patterns.

Response 11: Thank you for your suggestions.

Figure 7 has been revised Figure 6.

Figure 6 (a-e) The transverse energy flow of the controllable helico-conical beams with K = 0, l = 20, k = 0, 1, 4, 7, and 10, respectively. (f-j) The transverse energy flow of the controllable helico-conical beams with K = 1, l = 20, k = 0, 1, 4, 7, and 10, respectively.

Reviewer 3 Report

I think that the manuscript can be published in Photonics. The numerical calculations show many interesting behaviors of the Controllable helico-conical beam. The proposed helico-conical beams with the controllable length can be potentially applied in the field of optical guiding. However, some concerns are listed as follows which I encourage authors to further address before the final publication.

(1) The sentence of Page 2 “ Figure 1c and f......” can be changed into “Figures 1c and 1f......”

(2) The sentence of Page 4 “ it can be also seen that an......” can be changed into “ it is also can be seen that an......”

(3) The beam scales and colorbars in figures 2, 3 and 8 haven’t been indicated.

(4) In the process of writing, authors should increase these two references for  introducing the development of vortex.

[1]  Zhongsheng Man, Zheng Xi, Xiaocong Yuan, R. E. Burge, and H. Paul Urbach, Dual coaxial longitudinal polarization vortex structures, Physical Review Letters,   124(10): 103901 (2020).

[2] Peiwen Meng, Zhongsheng Man, A. P. Konijnenberg, Hendrik Paul Urbach, Angular momentum properties of hybrid cylindrical vector vortex beams in tightly focused optical systems, Optics Express 27(24):35336 (2019).

In all, I'm happy to recommend the manuscript "Controllable helico-conical beam generated with the bored phase " for publication in Photonics.

Author Response

Replies to Reviewer #3:

General Remarks: I think that the manuscript can be published in Photonics. The numerical calculations show many interesting behaviors of the Controllable helico-conical beam. The proposed helico-conical beams with the controllable length can be potentially applied in the field of optical guiding. However, some concerns are listed as follows which I encourage authors to further address before the final publication.

Response: Thank you for your appreciation of our contributions.

Comment 1: The sentence of Page 2 “Figure 1c and f.....” can be changed into “Figures 1c and 1f.....”

Response 1: Thanks for your suggestions.

The sentence of Page 2 “Figure 1c and f show the corresponding bored helico-conical phase with K = 0 and K = 1, respectively.” is revised as “Figures 1c and 1f show the corresponding bored helico-conical phase with K = 0 and K = 1, respectively.”.

Comment 2: The sentence of Page 4 “it can be also seen that an......” can be changed into “it also can be seen that an......”

Response 2: Thank you for your suggestions.

The sentence of Page 4 “From Figure 4b and Figure 4g, it can be also seen that an observable dislocation of the spiral’s head occurs.” is revised as “From Figure 4b and 4g, it also can be seen that an observable dislocation of the spiral’s head occurs.”.

Comment 3: The beam scales and color bars in figures 2, 3 and 8 haven’t been indicated.

Response 3: Thank you for your suggestions.

The color bar and scale bar are added into Figures 2, 3 and 8. The revised figures are as follows.

Figure 2. (a) The helico-conical phase profile with K = 0, l = 20. (b-e) The bored phase profiles with K = 0, l = 20, k = 1, 4, 7, and 10, respectively. (f-j) Simulated focal-field intensity distributions of the controllable helico-conical beams with K = 0, l = 20, k = 0, 1, 4, 7, and 10, respectively.

Figure 3. (a) The helico-conical phase profile with K = 1, l = 20. (b-e) The bored phase profiles with K = 1, l = 20, k = 1, 4, 7, and 10, respectively. (f-j) Simulated focal-field intensity distributions of the controllable helico-conical beams with K = 1, l = 20, k = 0, 1, 4, 7, and 10, respectively.

Figure 7. (a-e) The focal-field OAM distributions of the controllable helico-conical beams with K = 0, l = 20, k = 0, 1, 4, 7, and 10, respectively. (h-j) The focal-field OAM distributions of the controllable helico-conical beams with K = 1, l = 20, k = 0, 1, 4, 7, and 10, respectively.

Comment 4: In the process of writing, authors should increase these two references for introducing the development of vortex.

[1] Zhong sheng Man, Zheng Xi, Xiaocong Yuan, R. E. Burge, and H. Paul Urbach, Dual coaxial longitudinal polarization vortex structures, Physical Review Letters, 124(10): 103901 (2020).

[2] Peiwen Meng, Zhongsheng Man, A. P. Konijnenberg, Hendrik Paul Urbach, Angular momentum properties of hybrid cylindrical vector vortex beams in tightly focused optical systems, Optics Express 27(24):35336 (2019).

Response 4: Thanks for your suggestions.

New references cited as [12, 13] have been added to the manuscript. The reference list has also been reshuffled accordingly.

  1. Meng, P.W.; Man, Z.S.; Konijnenberg, A. P.; Urbach, H. P. Angular momentum properties of hybrid cylindrical vector vortex beams in tightly focused optical systems, Express 2019, 27(24), 35336.
  2. Man, Z.S.; Xi, Z.; Yuan, X.C.; Burge, R. E.; Urbach, H. P. Dual coaxial longitudinal polarization vortex structures, Phys. Rev. Lett. 2020, 124(10), 103901.

The sentence “The optical beams with the orbital angular momentum (OAM), e.g., optical vortices, have been widely used in optical communications and optical manipulation [1-11].” in lines 19-20 can be revised by “The optical beams with the orbital angular momentum (OAM), e.g., optical vortices, have been widely used in optical communications and optical manipulation [1-13].”.

Reviewer 4 Report

The paper addresses a quite reasonable subject and probably the results are correct.  However, the presentation is somewhat awkward, not only in the sense of the English language, which is far from good but generally understandable. I could not trace the unreasonably long algebra, the mathematical content of which is reduced to  calculation of an integral in terms of Bessel functions. It is not explained what is \rho and how it is related to the coordinates in which the integration is performed. J is not said to be the Bessel function, and no reference is given. Pictures are not good for understanding.  In particular I don't understand Fig 1c at all. The presentation is unnecessarily long and can be halved.

Language editing is needed throughout.

Author Response

Replies to Reviewer #4:

General Remarks: The paper addresses a quite reasonable subject and probably the results are correct.  However, the presentation is somewhat awkward, not only in the sense of the English language, which is far from good but generally understandable.

I could not trace the unreasonably long algebra, the mathematical content of which is reduced to calculation of an integral in terms of Bessel functions. It is not explained what is \rho and how it is related to the coordinates in which the integration is performed. J is not said to be the Bessel function, and no reference is given. Pictures are not good for understanding. In particular I don't understand Fig 1c at all. The presentation is unnecessarily long and can be halved.

Comments on the Quality of English Language

Language editing is needed throughout.

Response: Thank you for your suggestions.

The English has been polished.

The Figures 2, 3, 7, 8 have been modified. Figures 5 and 6 have been combined. The revised figures are as follows.

Figure 2. (a) The helico-conical phase profile with K = 0, l = 20. (b-e) The bored phase profiles with K = 0, l = 20, k = 1, 4, 7, and 10, respectively. (f-j) Simulated focal-field intensity distributions of the controllable helico-conical beams with K = 0, l = 20, k = 0, 1, 4, 7, and 10, respectively.

Figure 3. (a) The helico-conical phase profile with K = 1, l = 20. (b-e) The bored phase profiles with K = 1, l = 20, k = 1, 4, 7, and 10, respectively. (f-j) Simulated focal-field intensity distributions of the controllable helico-conical beams with K = 1, l = 20, k = 0, 1, 4, 7, and 10, respectively.

Figure 5 (a) The azimuth corresponding to the whole helico-conical beam is 3π/2. (b) The azimuth corresponding to the controllable helico-conical beam with K = 0, l = 20, and k = 12 is about 0.6712π. (c) Plots of the azimuth versus the parameter k of the controllable helico-conical beams with K = 0, l = 5, 10, 15, and 20, respectively. (d) Plots of the azimuth versus the parameter k of the controllable helico-conical beams with K = 1, l = 5, 10, 15, and 20, respectively.  The lines represent the linear regression lines.

Figure 6. (a-e) The transverse energy flow of the controllable helico-conical beams with K = 0, l = 20, k = 0, 1, 4, 7, and 10, respectively. (f-j) The transverse energy flow of the controllable helico-conical beams with K = 1, l = 20, k = 0, 1, 4, 7, and 10, respectively.

Figure 7. (a-e) The focal-field OAM distributions of the controllable helico-conical beams with K = 0, l = 20, k = 0, 1, 4, 7, and 10, respectively. (h-j) The focal-field OAM distributions of the controllable helico-conical beams with K = 1, l = 20, k = 0, 1, 4, 7, and 10, respectively.

In our manuscript there are not the above-mentioned  function \rho and J.

Figure 1(c) show the bored phases profile with l = 20, k = 6, K = 0. In fact, the phase profile shown in Figure 1(c) is the remaining one after a part of the whole helico-conical phase is bored.
